# Prevalence of and risk factors for hepatitis B and C viral co-infections in HIV infected children in Lagos, Nigeria

**Mary Adetola Lawal**[1,¤a]*, **Oluwafunmilayo Funke Adeniyi**[1,2☺], **Patricia Eyanya Akintan**[1,2☺], **Abideen Olurotimi Salako**[1,☺¤b], **Olorunfemi Sunday Omotosho**[3¤c‡], **Edamisan Olusoji Temiye**[1,2☺]

**1** Department of Paediatrics, Lagos University Teaching Hospital, Lagos, Nigeria, **2** Department of Paediatrics, College of Medicine, University of Lagos, Lagos, Nigeria, **3** Department of Community Health, Obafemi Awolowo University Teaching Hospital, Osun State, Nigeria

☺ These authors contributed equally to this work.
¤a Current address: Department of Paediatrics, Massey Street Children's Hospital, Lagos Island, Lagos, Nigeria
¤b Current address: GlaxoSmithKline, Ilupeju, Lagos, Nigeria
¤c Current address: Department of Clinical Sciences, Nigeria Institute of Medical Research, Yaba, Lagos, Nigeria
‡ These authors also contributed equally to this work.
* shalomti@yahoo.com

**Data Availability Statement:** All relevant data are within the manuscript and its Supporting information files.

## Abstract

### Introduction

The study was carried out to determine the prevalence of and risk factors for hepatitis B and C viral co-infections in HIV infected children in Lagos.

### Method

A cross-sectional study conducted to determine the prevalence and risk factors for hepatitis B and C viral co-infection in children aged 2 months to 13 years. There were 187 HIV infected and 187 HIV naïve age, sex-matched controls. Blood samples of participants were assayed for the serologic markers [HBsAg, anti-HBc, and anti-HCV)] of HBV and HCV viral infections using the Enzyme-Linked Immunosorbent assay (ELISA) method.

### Result

The prevalence of HBV infection using HBsAg was 5.3% and 4.8% (p = 0.814), among HIV-infected and HIV naïve children respectively, while using anti-HBc the prevalence was 7.0% and 7.5% (p = 0.842) among HIV- infected and HIV naïve children respectively. The prevalence of HCV infection among HIV- infected and HIV naive children were equal to 0.5% (p = 1.000). There was also no significant association with the identifiable risk factors (sharing of a toothbrush, sharing of needles, incision marks/tattoo, hepatitis B immunization status, history of blood transfusion, previous surgical operation, sexual exposure/abuse, history of jaundice, and genital circumcision) and the HBV and or HCV status among both groups of children. History of sexual exposure/abuse and history of jaundice were however found to

**Funding:** The author(s) received no specific funding for this work.

**Competing interests:** The authors have declared that no competing interests exist.

be predictors of the presence of HBsAg among HIV infected children only, using a binary logistic regression model.

## Conclusion

The prevalence of HBV and or HCV infection among HIV-infected children is similar to the prevalence among HIV naïve children, suggesting that HIV-infected children are not more predisposed to viral hepatitis than healthy children. Also, there was no significant difference in the prevalence of HBV infection irrespective of the use of HBsAg or anti-HBc.

## Introduction

Human Immunodeficiency Virus (HIV), hepatitis B virus (HBV), and hepatitis C virus (HCV) are the three most common chronic viral infections of public health importance with great socio-economic impact worldwide [1, 2]. Both HBV and HCV infections cause acute and chronic liver infection with the potential for liver cirrhosis and hepatocellular carcinoma (HCC) [3, 4].

Hepatitis C leads to chronic infection in 60–80% of patients after an acute infection [5, 6], while perinatal HBV infection is associated with a 90% risk of chronicity compared with a risk of less than 5% among adults with intact immunity [7].

Globally, over 350 million and 170 million people are chronically infected with HBV and HCV respectively [4, 8]. In Africa, about 100 million individuals are estimated to be infected with HBV or HCV [9]. Also, HBV and HCV infections are highly endemic in Africa and are responsible for 80% of liver cirrhosis and HCC, with HBV being the main cause of end-stage liver disease [9].

Human Immunodeficiency Virus (HIV) destroys and impairs the functions of the immune cells of individuals who become immune-deficient predisposing them to acute, chronic infections, chronic disease, malignancy, and end-organ damages [10, 11]. HIV-infected people are three to six times more likely to develop chronic or long-term hepatitis B infection because of their suppressed immune systems than individuals without HIV [12]. Studies [1, 4–6, 13] show that HIV adversely impacts the natural history of HBV and HCV by accelerating progression to chronic liver disease (CLD) due to drug-related hepatotoxicity and hepatitis reactivation, resulting in high chances of liver-related diseases compared to individuals with HIV infection alone and vice versa. The three viruses share similar modes of transmission, common risk factors despite their biological differences, and hence coexist in the same host at a significantly high rate [5, 6, 13]. However, several factors such as geographical region, risk groups, age of infection, modes of transmission efficacy, or efficiency of exposure can influence and modify the prevalence [14].

The global prevalence of HBV/HIV co-infection varies from 1.13% to 59% [15]. In the United States of America [USA], the prevalence of HIV/HBV in children is 2.6% and 4.9% in China [16, 17]. Reports from Africa have revealed that the prevalence of HBV/HIV co-infection is between 10% and 20% as many countries in sub-Saharan Africa are typically classified as endemic, high, or intermediate countries with HBV infections [18]. In Tanzania, a prevalence of 1.2% was documented in children aged 18 months to 17 years while 12.1% was documented by Route et al in Cote d' Ivoire in West Africa [19, 20].

The global prevalence of HCV/HIV co-infection is heterogeneous but lower than that of the HBV/HIV co-infection. The prevalence of HCV infection in a large cohort of children perinatally infected with HIV in a long-term follow-up protocol in the United States is 1.5% while a prevalence rate of 9.6% was reported by Zhou et al in China [17, 21]. A higher prevalence of this co-infection, however, has been observed in some Africa countries like Tanzanian where a prevalence of 13.8% has been documented in children, on the other hand, a prevalence

rate of 0% was obtained by Route et al in Cote d' Ivoire in West Africa [19, 20]. Similarly, a study in East Africa reported 0% for the prevalence of HCV/HIV co-infection [22].

However, in Nigeria with a high burden of HIV, HBV, and HCV infection, most studies on co-infections are predominantly adult studies [2, 23–25] and there are very few paediatric studies [5, 6, 26, 27] that have documented the HBV/HIV co-infection or HCV/HIV co-infection status in Nigeria children despite the high impact of these viral infections on children in the Nigerian environment. The knowledge of the pattern of co-infection of HBV and HCV in children with HIV/AIDS enables prompt management of such children which can help to reduce associated morbidity, mortality and ensures continued consolidation of the gains of HAART on HIV/AIDS care.

Thus, this study was carried out to determine the prevalence of and risk factors for hepatitis B and C viral co-infection in HIV infected children in Lagos. The aim of the study was achieved.

## Materials and methods

We conducted a cross-sectional study at the Lagos University Teaching Hospital, Lagos (LUTH), in South-west, Nigeria, from August 2017 to December 2017. LUTH is a tertiary hospital that serves as a referral centre providing healthcare services to the State and its environs. Recruitment of study participants was done at the HIV clinic, well-baby, and the Paediatric outpatient clinics using the systematic sampling method.

### Study population

The study participants were children with a confirmed diagnosis of HIV infection aged 2 months to 13 years attending the Paediatric HIV clinic of LUTH.

The presence of HIV was based on documentary evidence of a positive DNA/PCR for children less than 18 months and positive results following the serial algorithm for HIV screening in children greater than 18 months as shown in Fig 1. While the HIV naive (controls) were healthy children attending the Well Baby Clinic and the paediatric outpatient unit with no evidence of HIV infection on a rapid screening test. These children were matched by age and sex to the subjects.

### Study criteria

### Inclusion criteria for subjects (HIV infected)

1. Children aged between 2 months to 13 years who are HIV positive following the HIV screening algorithm.

2. Children whose parents/guardians gave informed consent to participate in the study; (and assent obtained for children 7years and above).

3. Those that completed the administered questionnaire properly.

### Exclusion criteria for subjects (HIV infected)

- Children whose parent(s) / guardian refused or withdrew their consent.

- Those with acute febrile illness, seizure disorders, and cerebral palsy.

### Inclusion criteria for HIV naive (healthy controls)

1. Children aged between 2 months to 13 years with no evidence of HIV infection on a rapid screening test.

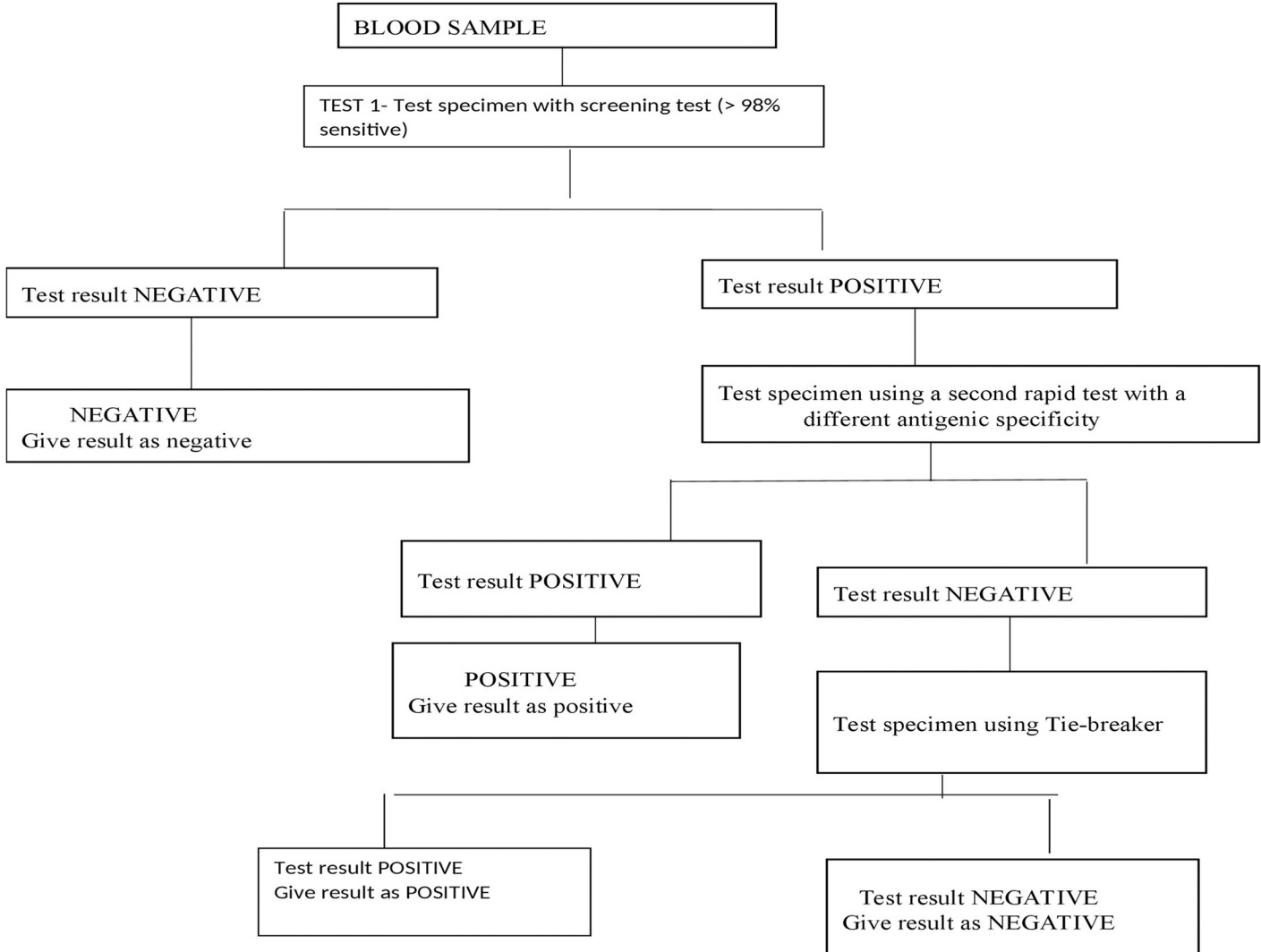

**Fig 1. Serial algorithm for rapid HIV testing.** [28] SSS These are WHO certified rapid kits use in LUTH. Test 1 (Determine, the sensitivity of 100% and specificity of 99.9%), test 2 (Unigold, the sensitivity of 100% and specificity of 99.9%), and test 3 (statpak which is a tiebreaker with 100% sensitivity and 100% sensitivity).

2. Children whose parents/guardians gave informed consent to participate in the study; (and assent obtained for children 7years and above).

3. Participants who completed the administered questionnaire properly.

## Exclusion criteria for HIV naive (healthy controls)

- Children whose parent(s) / guardians refused or withdrew their consent.

- Children with chronic diseases such as confirmed sickle cell anaemia, confirmed chronic liver disease, cerebral palsy, and seizure disorders.

## Sample size determinations

The minimum sample size was determined using the formula for calculating sample size for the comparison of two independent proportions [29].

$$n \text{ per group} = \frac{2 \left(Z\alpha + Z_\beta\right)^2 P \left(1 - P\right)}{\left(Po - P_1\right)^2}$$

Where

$Z\alpha$ = the standard normal deviate of $\alpha$ at 95% confidence interval, (the probability of making a type 1 error) = 1.96

$Z_\beta$ = the standard normal deviate of $\beta$ (the power) is set at 90% = 1.28

$Po$ = the prevalence of hepatitis B co-infection among HIV-infected children

A prevalence of 10% was used based on the work by Durowaye et al [30].

$P_1$ = the prevalence of hepatitis B in children;

A prevalence of 0.5% was used based on the work by Okonko et al [31].

$P$ is the arithmetic average of the two proportions $\frac{Po+P_1}{2}$

## Data collection

A pre-tested proforma was administered to obtain biodata, clinical information, and socio-demographic characteristics (age, gender, ethnicity, the socioeconomic class which includes occupational status and the highest level of education attained by each parent and or guardian using Oyedeji classification of social class [32]). Immunization history, presence of risk factors for transmission of HBV and HCV such as incision marks on the skin, previous history of blood transfusion, unsafe injections, previous surgery and genital circumcision, and the presence of clinical features of hepatitis (jaundice) in the preceding 6 months before recruitment into the study were also obtained. Immunization cards of all those who received the vaccine were demanded and verified.

Genital circumcision is defined as the removal of the foreskin from the penis in this manuscript.

Venepuncture for sample collection was done under strict asepsis in the procedure room of the paediatric outpatient clinic and HIV clinic.

Two millilitres of blood were obtained per participants aseptically from a peripheral vein into an Ethylene-Diamine-Tetra-Acetic Acid (EDTA) specimen bottles for HBV and HCV screening. Thereafter the samples were transported in ice to the Central Research Laboratory of the College of Medicine University of Lagos, [LUTH/CMUL] Idi Araba Lagos within 24hrs after collection where it was centrifuged at 3,000rpm for five minutes to separate the plasma. The plasma was extracted using micropipette into 1.25ml screw cap tubes and stored at -200C until analysis was performed.

## Laboratory analysis

Hepatitis B virus infection was confirmed using Monolisa™HBsAg ULTRA assay (4[th] generation enzyme immunoassay; BIO-RAD Laboratories France). This kit has a sensitivity of 100% and a specificity of 99.94%. It is a qualitative one-step enzyme immunoassay technique of the "sandwich" type for the detection of the surface antigen of the Hepatitis B virus (HBsAg) in the human serum or plasma.

Antibody to HBV core antigen (anti-HBc) was screened for using (Monolisa™ Anti-HBc PLUS; BIO-RAD Laboratories, France). It is an enzyme immunoassay (indirect ELISA type) for the simultaneous detection of total antibodies to hepatitis B virus core in human serum or

plasma. The detection of antibodies to hepatitis B nucleocapsid or core antigen is a major marker for the presence of past (anti-HBc Total) or recent (anti-HBc IgM) infection.

Antibody to HCV antigen (anti-HCV) was screened for using (Version 4.0 Enzyme Immunoassay; DIA.PRO Diagnostic Bioprobes Srl, Italy.) It is an indirect qualitative enzyme immunoassay for the detection of anti-Hepatitis C virus antibody in human serum and plasma.

## Statistical analysis

Data were analysed using the Statistical Package for the Social Sciences (SPSS) for Windows version 20. Continuous data were tested for distribution and the mean and standard deviation were calculated. Continuous data and categorical data were compared using the Student t-test, chi-square test, and Fischer's exact test where applicable. The prevalence of HBV and HCV among respondents was calculated using proportions. Pearson's Chi-Square (or Fisher Exact test where necessary) was used to compare the prevalence of HBV and HCV between HIV-positive and HIV-negative respondents. A Chi-square test was used to assess the relationship between different risk factors and HBV and or HCV infection among respondents. Binary logistic regression analysis was also used to determine factors predictive of the infection. Probability (p) values less than 0.05 were accepted as statistically significant.

## Ethical approval

Ethical approval was obtained from LUTH Health Research Ethics Committee before the commencement of the study. All the parents/ guardian and the patients present on each clinic day were addressed and the primary investigator subsequently had a one on one communication with the selected parents /guardian and the patients, where details of the study were communicated to them. Thereafter, written informed consent was obtained from each child's parent or guardian before study entry, and informed assent was obtained from each child aged seven years or more.

## Results

### General characteristics

A total of three hundred and seventy-four (374) children were enrolled in the study, comprising one hundred and eighty-seven (187) HIV-infected and HIV naïve (healthy controls) respectively. The mean ± SD age of HIV infected participants was 8.54 ± 3.29 years and that of the HIV naïve (healthy controls) was 8.01 ± 3.40 years. The male to female ratio in both groups was 1.1:1. The majority (45.5%) of the study participants were in the age group 10 to 13 years while 3.2% were infants. Also, the majority (51.3%) of the study participants were males as shown in Table 1. There was no significant difference between HIV infected children and the HIV naïve (control) subjects for age and sex (p = 1.000).

There was a significant difference in the socio-economic status between the two groups. Among the HIV infected subjects, 119 (63.6%) were in the middle socio-economic class compared with 98 (52.4%) among HIV naïve while 22 (11.7%) were in lower socio-economic class compared with 15 (8.0%) among the HIV naïve, (p = 0.009) as shown in Table 1. There was also a significant difference in the tribe between the two groups. The majority 74 (39.3%) of HIV-infected children belonged to the Igbo tribe compared with 98 (59.0%) among HIV naïve (controls) who were of the Yoruba tribe. 3 (1.6%) were of the Hausa tribe among HIV-infected while 6 (3.2%) were among HIV naive children (p = 0.000) as shown in Table 1.

The prevalence of HBV infection among HIV infected children (using HBsAg) was 5.3% compared with a rate of 4.8% among HIV naïve (healthy controls) children (p = 0.814), while

**Table 1. Socio-demographic characteristics of study participants.**

| Socio-demographic characteristics | HIV infected n (%) | HIV naïve(controls) n (%) | Test of Statistics $\chi^2$ | P-Value |
|---|---|---|---|---|
| Age | | | | |
| < 1 year | 6(3.2%) | 6(3.2%) | | |
| 1–5 years | 29(15.5%) | 29(15.5%) | | |
| 6–9 years | 67(35.8) | 67(35.8) | 0.000 | 1.000 |
| 10–13 years | 85(45.5) | 85(45.5) | | |
| Sex | | | | |
| Male | 96(51.3) | 96(51.3) | 0.000 | 1.000 |
| Female | 91(48.7) | 91(48.7) | | |
| SE status | | | | |
| Upper | 46(24.6) | 74(39.6) | | |
| Middle | 119(63.6) | 98(52.4) | 13.624 | 0.009* |
| Lower | 22(11.7) | 15(8.0) | | |
| Tribe | | | | |
| Yoruba | 68(36.4) | 98(59.0) | | |
| Hausa | 3(1.6) | 6(3.2) | 18.424 | 0.000* |
| Igbo | 74(39.3) | 67(35.8) | | |
| Other | 42(22.5) | 16(8.6) | | |

SE: Socioeconomic class,

* = significant p-value, Other (Ijaw, Efik, Tiv).

the prevalence of HBV infection (using anti-HBc) was 7.0% among HIV infected subject compared with a rate of 7.5% among HIV naïve (p = 0.842). There were no significant differences in the prevalence of HBV infection irrespective of the use of HBsAg or anti-HBc as shown in Table 2.

The prevalence of HCV infection among HIV-infected children was 0.5% which was comparable with a rate of 0.5% among HIV naïve (p = 1.000). There were no significant differences in the prevalence of HCV between the two groups studied as shown in Table 2.

The gender distribution of HBV among HIV infected was of equal distribution; 5 (5.2%) males and 5 (5.5%) females tested positive for HBsAg. The only subject with HIV/HCV co-infection was a male. None of the HIV infected children was co-infected with all three viruses. Among HIV naïve children 5 (5.2%) males and 4 (4.4%) females tested positive for HBsAg respectively. The only subjects with HCV infection (anti-HCV positive) was a female.

**Table 2. Comparison of the prevalence of HBV and or HCV viral co-infections among HIV-infected and HIV naïve (controls) children.**

| Variables | Results | HIV infected n = 187(%) | HIV naïve (control) n = 187(%) | Total n = 374(%) | Statistics $\chi^2$ | P-value |
|---|---|---|---|---|---|---|
| HBsAg | Positive | 10(5.3) | 9(4.8) | 19(5.1) | 0.055 | 0.814 |
| | Negative | 177(94.7) | 178(95.2) | 355(94.9) | | |
| Anti-HBc | Positive | 13(7.0) | 14(7.5) | 27(7.2) | 0.040 | 0.842 |
| | Negative | 174(93.1) | 173(92.5) | 347(92.8) | | |
| Anti-HCV | Positive | 1(0.54) | 1(0.54) | 2(0.5) | Fisher's | |
| | Negative | 186(99.5) | 186(99.5) | 372(99.5) | Exact = 0.751 | 1.000 |

HIV: Human immunodeficiency virus, HBsAg: Hepatitis B surface antigen, Anti-HBc: Antibody to hepatitis B virus core antigen, Anti-HCV: Antibody to hepatitis C virus.

There was no significant association between the risk factors and the HBV and HCV status of the HIV-infected and the HIV naïve children as shown in Tables 3 and 4.

Of all the variables put in the model, only history of sexual exposure/ abuse (p = 0.041) and history of jaundice (p = 0.032) were found to be predictors of the presence of HBsAg and this was only among HIV positive participants, while it was not so among HIV naïve children as shown in Table 5.

## Discussion

The prevalence of HBV/HIV co-infection obtained in this study is 5.3% and is comparable to the 5.8%, 5.8%, and 6.02% reported in South-east and South-south, Nigeria, and 4.9% in China [17, 26, 27, 33]. The prevalence rate obtained in the index study could be attributed to various

**Table 3. Comparison of the risk factors for HBV acquisition among HIV infected and HIV naïve children.**

| Variables | HIV infected HBsAg +n = 10(%) | HIV infected HBsAg- n = 177(%) | StatisticsFisher Exact/ P-value | HIV naïve HBsAg + n = 9(%) | HIV naïve HBsAg- n = 178(%) | StatisticsFisher Exact/ P-value |
|---|---|---|---|---|---|---|
| Blood transfusion | | | | | | |
| Yes | 1(10) | 41(23.2) | 0.296 | 1(11.1) | 40(22.5) | 0.371 |
| No | 9(90) | 136(76.8) | 0.460 | 8(88.9) | 138(77.5) | 0.686 |
| Incision marks/ tattoo | | | | | | |
| Yes | 0(0) | 16(9.0) | 0.393 | 0(0) | 13(7.3) | 0.515 |
| No | 10(100) | 161(91.0) | 0.605 | 9(100) | 165(92.7) | 1.000 |
| Hepatitis B vaccination | | | | | | |
| Yes | 6(60) | 93(52.5) | 0.457 | 9(100) | 130(73) | 0.071 |
| No | 4(40) | 84(47.5) | 0.752 | 0(0) | 48(27) | 0.115 |
| Sharing of needles | | | | | | |
| Yes | 0(0) | 2(1.1) | 0.896 | 0(0) | 2(1.1) | 0.905 |
| No | 10(100) | 175(98.9) | 1.000 | 9(100) | 176(98.9) | 1.000 |
| Sharing of toothbrush | | | | | | |
| Yes | 0(0) | 2(1.1) | 0.896 | 0(0) | 13(7.8) | 0.515 |
| No | 10(100) | 175(98.9) | 1.000 | 9(100) | 178(92.7) | 1.000 |
| Past surgical operation | | | | | | |
| Yes | 1(10) | 3(1.7) | 0.199 | 0(0) | 8(4.5) | 1.000 |
| No | 9(90) | 174(98.3) | 0.199 | 9(100) | 170(95.5) | 0.669 |
| Genital circumcision | | | | | | |
| Yes | 4(40) | 89(50.3) | 0.747 | 5(55.6) | 86(48.3) | 0.742 |
| No | 6(60) | 88(49.7) | 0.380 | 4(44.4) | 92(51.7) | 0.466 |
| Sexual exposure/ abuse | | | | | | |
| Yes | 1(10) | 2(1.1) | 0.153 | 0(0) | 0(0) | 1.000 |
| No | 9(90) | 175(98.9) | 0.153 | 9(100) | 178(100) | 1.000 |
| Past history of jaundice | | | | | | |
| Yes | 2(20) | 6(3.4) | 0.061 | 1(11.1) | 24(13.5) | 0.653 |
| No | 8(80) | 171(96.6) | 0.061 | 8(88.9) | 154(86.5) | 1.000 |

HIV: Human immunodeficiency virus, HBsAg+: Hepatitis B surface antigen-positive, HBsAg⁻: Hepatitis B surface antigen-negative.

**Table 4. Comparison of the risk factors for HCV acquisition among HIV infected and HIV naive children.**

| Variables | HIV infected Anti-HCV+n = 1 (%) | HIV infected Anti HCV-n = 186 (%) | Statistics Fisher's Exact/P value | HIV naive Anti-HCV-n = 1(%) | HIV naive Anti-HCV-n = 186 (%) | Statistics Fisher's Exact/ P value |
|---|---|---|---|---|---|---|
| Blood transfusion | | | | | | |
| Yes | 0 (0) | 42 (22.6) | 0.774 | 0 (0) | 41 (22) | 0.780 |
| No | 1 (100) | 144 (77.4) | 1.000 | 1 (100) | 145 (78) | 1.000 |
| Incision marks/tattoo | | | | | | |
| Yes | 0 (0) | 16 (8.6) | 0.913 | 0 (0) | 13 (7.0) | 0.930 |
| No | 1 (100) | 170 (91.4) | 1.000 | 1 (100) | 173 (93.0) | 1.000 |
| Hepatitis B vaccination | | | | | | |
| Yes | 1 (100) | 98 (52.7) | 0.532 | 1 (100) | 138 (74.2) | 0.751 |
| No | 0 (0) | 88 (47.3) | 1.000 | 0 (0) | 48 (25.8) | 1.000 |
| Sharing of needles | | | | | | |
| Yes | 0 (0) | 2 (1.1) | 0.989 | 0 (0) | 2 (1.1) | 0.989 |
| No | 1 (100) | 184 (98.9) | 1.000 | 1 (100) | 184 (98.9) | 1.000 |
| Sharing of toothbrush | | | | | | |
| Yes | 0(0) | 2(1.1) | 0.989 | 0(0) | 13(7.3) | 0.930 |
| No | 1(100) | 184(98.9) | 1.000 | 1(100) | 173(92.7) | 1.000 |
| Past surgical operation | | | | | | |
| Yes | 0(0) | 4(2.2) | 0.979 | 0(0) | 8(4.3) | 0.957 |
| No | 1(100) | 182(97.8) | 1.000 | 1(1.000) | 178(95.7) | 1.000 |
| Genital circumcision | | | | | | |
| Yes | 1(100) | 92(49.5) | 0.497 | 0(0) | 91(48.9) | 0.513 |
| No | 0(0) | 94(50.5) | 1.000 | 1(100) | 95(51.1) | 1.000 |
| Sexual exposure/abuse | | | | | | |
| Yes | 0(0) | 3(1.6) | 0.984 | 0(0) | 0(0) | 1.000 |
| No | 1(100) | 183(98.4) | 1.000 | 1(100) | 186(100) | 1.000 |
| Past history of jaundice | | | | | | |
| Yes | 0(0) | 8(4.3) | 0.957 | 0(0) | 25(13.4) | 0.866 |
| No | 1(100) | 178(95.7) | 1.000 | 1(100) | 161(86.6) | 1.000 |

Anit-HCV+: Antibody to hepatitis C virus-positive, Anti-HCV−: Antibody to hepatitis C virus-negative.

reasons which include the wide coverage of the hepatitis B vaccine in universal vaccination programs in our country and worldwide; thus ensuring substantial progress towards eliminating hepatitis B infection in children [34, 35]. The increase in hepatitis B vaccination in Nigeria from 42% to approximately 50% in the last decade [35, 36]. may have also contributed to this finding. Other plausible reasons for our observation may also be the prompt commencement of highly active antiretroviral therapy (HAART) in the HIV-infected children and increased awareness on healthy lifestyles and the preventive measures of the HBV and or HIV-infected individuals, family, and caregivers.

However, the prevalence of HBV/HIV co-infection in the current study was lower than 19% documented in Northern Nigeria [37] and the reason for the variation may be attributed to the fact that the study participants in the present study were all on HAART and had access

**Table 5. Binary logistic regression showing the adjusted odds ratio of the determinants of hepatitis B infection among HIV infected and HIV naïve children.**

| HIV positive | | | | HIV negative | | |
|---|---|---|---|---|---|---|
| Variable | Odds ratio | CI | P-value | Odds ratio | CI | P-value |
| Previous blood transfusion | 3.803 | 0.381–37.969 | 0.255 | 2.023 | 0.228–17.969 | 0.527 |
| History of incision marks/tattoos | 1355.4 | 0–0 | 0.998 | 6594.5 | 0–0 | 0.999 |
| Practice sharing of needles | 7304.5 | 0–0 | 0.999 | 0.847 | 0–0 | 1.000 |
| Practice sharing of toothbrush | 3373.6 | 0–0 | 0.999 | 5072.7 | 0–0 | 0.999 |
| History of surgical operation | 0.087 | 0.007–1.163 | 0.065 | 9711.5 | 0–0 | 0.999 |
| History of genital circumcision | 1.658 | 0.338–8.138 | 0.533 | 0.729 | 0.184–2.883 | 0.652 |
| History of sexual exposure/abuse | 0.062 | 0.004–0.893 | 0.041* | 0.796 | 0.086–7.352 | 0.841 |
| History of jaundice | 0.110 | 0.015–0.83 | 0.032* | 0.796 | 0.086–7.352 | 0.841 |

CI = 95% confidence interval,

* = significant p value,

HIV: Human immunodeficiency virus

to Hepatitis B vaccine amongst other vaccines available in the NPI schedule. On the other hand, the study participants in the study from Northern Nigeria were a HAART naive cohort. Other reports on HAART naïve study participants from Makurdi [38], Ilorin [30] and Jos [39] cities in Northcentral Nigeria and Ivory Coast [20] another country in West Africa have also reported a higher prevalence of 7.8%, 10%, 12.8%, 12.1% respectively in the cohorts studied compared to the prevalence obtained in this present study. The difference in the prevalence in these studies compared to the current study may be attributed to the differences in sample sizes, differences in the age of the study population as some of the studies included adolescents, and possibly the variable geographical distribution of HBV infection in the area of study [6].

Nevertheless, the lower prevalence obtained in the current study suggests and possibly affirms the benefit of HAART on inhibition of HBV/ HIV replication.

The prevalence of HBV infection in HIV naïve children in this study was 4.8% and 7.5% using HBsAg and anti-HBc respectively. This prevalence falls within the HBsAg prevalence range of 2–7% for intermediate endemicity for HBV infection, although Nigeria and most parts of sub-Saharan Africa fall in the high endemic zone with a prevalence of >8% [40].

However, the prevalence rate of 4.8% for HBV infection (HBsAg) among the HIV naïve participants is in keeping with previously documented prevalence rates of 3.1%, 4.3%, and 3.9% in Southern Nigeria [41–43]. This study including the aforementioned ones was reported after the implementation of HBV vaccination and thus the finding could be attributable to the success of the inclusion of hepatitis B vaccination into the national program on immunization (NPI) schedule in our country, buttressing the fact that HBV infection (both acute and chronic infection) has been on the downward trend in Nigeria [42, 44].

However, the prevalence rate obtained in the controls (HIV naïve children) in our study was higher than those reported from some cities in Southern Nigeria 1.2% [31] and 0.5% [45] but lower than 13.9% in Benin, Southern Nigeria, by Sadoh *et al* [46] and a rate of 8.0% in Maiduguri, Northern Nigeria, reported by Isa *et al* [47]. This difference in prevalence also affirms the variations in the occurrence of HBV infection within Nigeria and beyond and this could also be attributed to varying cultural belief, religious beliefs, and health-seeking behaviour.

Additionally, the variation in the prevalence of HBV/HIV co-infection across the country further buttresses the difference in Hepatitis B vaccine coverage across the country. The HBV coverage in the Southern part of Nigeria where the current study was undertaken is between 74.8–90.6% compared to the Northern part where the HBV coverage ranges from 29.1%-74% [36].

There is no statistically significant difference in HBV prevalence between the HIV-infected and HIV naïve children in the present study, in agreement with a previous report in South Africa by Mdlalose *et al* [48] whose study design was similar to the present study. This finding is in concordance with reports by past authors who compared the prevalence of HBV between HIV-infected children and HIV negative children [6, 30]. This suggests that the prevalence of HBV infection in children may be independent of HIV and a reflection of the ongoing success of HBV vaccination in the NPI schedule in Nigeria and globally.

However, contrary to our findings of no significant difference in the prevalence rate of HBV among HIV-infected compared with HIV naïve children Okeke *et al* [33] in Nnewi, Southern Nigeria, reported a higher prevalence rate of HBV among HIV infected compared to HIV naïve children. Okeke et al alluded the findings to the common mode of transmission of the diseases, the decrease rate of HBsAg clearance in HIV positive children because of the associated immune suppression [30].

The HCV/HIV co-infection prevalence rate of 0.5% in our study is comparable to zero prevalence rates in Ivory Coast and Kenya [20, 49]. The low prevalence rate obtained in this study reflects the low prevalence of HCV in the Nigerian general population among children as documented by previous researchers [50–53]. The robust and extensive national and stable blood transfusion program, which enforces mandatory screening for HCV and among other transfusion transmittable infections may explain the observed low prevalence of hepatitis C among HIV-infected children [54]. Also, the low prevalence could be due to the use of a more sensitive $4^{th}$ generation enzyme-linked immunosorbent assay in this study, and also it could be because of the expanded use of HAART in HIV-infected individuals.

In contrast to the low prevalence rate of 0.5% for HCV/HIV co-infection in this study, a higher prevalence rate of 1.7% was reported by Durowaye *et al* [30] in Northcentral, Nigeria. Furthermore, a higher prevalence of 1.5% was also reported by Schuval *et al* [21] in the USA, although a more sensitive screening method HCV RNA by RT-PCR was adopted to diagnose HCV at a lower viral load of 50 copies/ ml which could not be applied due to cost in the index study. Other workers in Nigeria like Sadoh *et al* [6] and Ejeliogu *et al* [5] have also reported higher prevalence rates of 5.2% and 5.0% in Benin and Jos respectively. Possible reasons for these findings may also be due to the mixed population studied, by Sadoh *et al* [6] where the adolescents accounted for a higher population of HCV/HIV co-infection. The participants studied by Ejeliogu *et al* [5] were all HAART naïve and this may also possibly explain the high prevalence of co-infection observed in the cohort.

The prevalence rate of 0.5% for HCV among HIV naïve in the current study is comparable to prevalence reports in Nigeria and around the world [41, 50–53, 55]. This finding further buttresses the low prevalence rate of HCV in children and the general population.

Meanwhile, a higher prevalence rate of 10% was reported in healthy children in Northern Nigeria, by Isa *et al* [56] compared with the prevalence in this study. This could also be alluded to the variation in the distribution of HCV infection, in addition to the fact that the sample size in the previous study was small.

There is also no significant difference in the HCV infection among the HIV-infected compared with HIV naïve children. However, there is a paucity of works that directly compared the prevalence of HCV in HIV infected compared to HIV naïve, though existing works have shown the low prevalence in the two study groups [51–53, 57]. Thus, this is explainable by the low prevalence of HCV in the general population.

The investigated risk factors such as unsafe injection practices, sharing of sharp objects, genital circumcision, blood transfusion, sexual exposure/abuse, previous surgical operation, and scarification/tattoo in the current study showed no significant association with the prevalence of HBsAg and HCV infection in either HIV-infected or HIV naïve children. This is in

concordance with a previous report by earlier researchers [6, 27, 42, 58]. The rationale behind these findings could be due to current ongoing awareness and adoption of healthy lifestyles, preventive measures by the population, the robust national blood transfusion program, and better access to health care services. However, in contrast to the findings in the present study other reports have observed association with the highlighted risk factors the occurrence of HCV and HBV in HIV-infected or HIV naïve children [33, 37, 43].

Sexual exposure/abuse and past history of jaundice were predictors of the presence of HBsAg among HIV infected subjects only in this study on the binary logistic model. It must be noted, however, that sexual transmission of hepatitis B is relatively uncommon in the paediatric age group as observed in this study.

This study is limited by the fact that the HBV detection test was done at a single contact and thus the possibility of chronic HBV infection could not be confirmed. Additionally, HCV reverse transcriptase-polymerase chain reaction (RT-PCR) which is a more sensitive diagnostic test for HCV infection could not be done for financial and logistic reasons for the study participants, which is the most sensitive test currently available for HCV diagnosis.

In conclusion, the prevalence of HBV and or HCV infection in HIV-infected children is similar to the prevalence of HBV or HCV infection among HIV naïve children and the general population, suggesting that HIV-infected children are not more predisposed to viral hepatitis than healthy children in the general population. Also, there was no significant difference in the prevalence of HBV infection irrespective of the use of HBsAg or anti-HBc. Nevertheless, the screening of HBV and HCV infection in HIV infected children is still recommended to give appropriate treatment and prevent accelerated liver disease and improve prognosis in this group of children.

## Supporting information

**S1 Appendix. Questionnaire.**
(DOCX)

**S1 File.**
(XLSX)

## Acknowledgments

With all humility, I sincerely appreciate the children and the parents or guardians who participated in the study. I am also immensely grateful to my fellow residents, colleagues, and the entire staff of LUTH/CMUL Research Laboratory for their support and kind words of encouragement.

## Author Contributions

**Conceptualization:** Mary Adetola Lawal, Oluwafunmilayo Funke Adeniyi, Patricia Eyanya Akintan.

**Data curation:** Mary Adetola Lawal, Patricia Eyanya Akintan, Abideen Olurotimi Salako.

**Formal analysis:** Mary Adetola Lawal, Olorunfemi Sunday Omotosho.

**Funding acquisition:** Mary Adetola Lawal, Oluwafunmilayo Funke Adeniyi, Patricia Eyanya Akintan, Olorunfemi Sunday Omotosho, Edamisan Olusoji Temiye.

**Investigation:** Mary Adetola Lawal, Patricia Eyanya Akintan, Edamisan Olusoji Temiye.

**Methodology:** Mary Adetola Lawal, Patricia Eyanya Akintan, Abideen Olurotimi Salako, Edamisan Olusoji Temiye.

**Project administration:** Mary Adetola Lawal, Edamisan Olusoji Temiye.

**Resources:** Mary Adetola Lawal, Abideen Olurotimi Salako, Olorunfemi Sunday Omotosho, Edamisan Olusoji Temiye.

**Software:** Olorunfemi Sunday Omotosho.

**Supervision:** Mary Adetola Lawal, Oluwafunmilayo Funke Adeniyi, Patricia Eyanya Akintan, Abideen Olurotimi Salako, Edamisan Olusoji Temiye.

**Validation:** Mary Adetola Lawal, Oluwafunmilayo Funke Adeniyi, Edamisan Olusoji Temiye.

**Visualization:** Mary Adetola Lawal, Patricia Eyanya Akintan, Abideen Olurotimi Salako.

**Writing – original draft:** Mary Adetola Lawal.

**Writing – review & editing:** Mary Adetola Lawal, Oluwafunmilayo Funke Adeniyi, Patricia Eyanya Akintan, Abideen Olurotimi Salako, Edamisan Olusoji Temiye.

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
