## [Decision Letter · Decision Letter 0]

16 Sep 2020

PONE-D-20-23898

Prevalence of and risk factors for hepatitis B and C viral co-infections in HIV infected children in Lagos, Nigeria.

PLOS ONE

Dear Dr. Lawal,

Thank you for submitting your manuscript to PLOS ONE. After careful consideration, we feel that it has merit but does not fully meet PLOS ONE’s publication criteria as it currently stands. Therefore, we invite you to submit a revised version of the manuscript that addresses the points raised during the review process.The overall writting is confusing and is difficult to follow-up. Probably due to this it is difficult to compare those results to others in the litterature. It would be improved following the review comments and results better put in perspective.

We look forward to receiving your revised manuscript.

Kind regards,

Isabelle Chemin, PhD

Academic Editor

PLOS ONE

Journal Requirements:

2. In your Methods section, please provide additional information about the participant recruitment method and the demographic details of your participants. Please ensure you have provided sufficient details to replicate the analyses such as:

a) the recruitment date range (month and year),

b) a description of any inclusion/exclusion criteria that were applied to participant recruitment,

c) a statement as to whether your sample can be considered representative of a larger population, and

d) a description of how participants were recruited.

3. We note that you use the term "genital circumcision" in your manuscript but do not define this. Please define if this includes removal of the foreskin of the penis, removal of the visible part of the clitoris, or other practices.

4. Please provide the name and catalog number of the anti-HCV Version 4.0 Immunoassay.

Reviewers' comments:

Reviewer's Responses to Questions

**Comments to the Author**

1. Is the manuscript technically sound, and do the data support the conclusions?

Reviewer #1: Partly

2. Has the statistical analysis been performed appropriately and rigorously? 

Reviewer #1: Yes

3. Have the authors made all data underlying the findings in their manuscript fully available?

Reviewer #1: Yes

4. Is the manuscript presented in an intelligible fashion and written in standard English?

Reviewer #1: No

5. Review Comments to the Author

Reviewer #1: 1. When describing prevalence of HIV/HBV and HIV/HBV respectively, especially in the Introduction when describing the published literature, please be clearer and describe the range prevalence for each of the viral co-infections. In the current form, it is rather unclear what the authors are referring to in their ranges of prevalence.

2. In the discussion, more can and should be said about whether there are variations in the rate of Hepatitis B vaccination between the different regions in Nigeria in which the prevalence of HIV/HBV has been studied, to account for the differences reported.

3. The discussion is rather confusingly structured. Would recommend that the authors reframe it speaking of each of the viral hepatitides in sequence, without the current structure which shifts between HBV and HCV, for ease of understanding.

4. A thorough proofreading and rewriting where needed to make it easier to read and understand would also help in the overall quality of the manuscript.

6. PLOS authors have the option to publish the peer review history of their article (what does this mean?). If published, this will include your full peer review and any attached files.

Reviewer #1: No

---

## [Author Response · Author response to Decision Letter 0]

5 Nov 2020

Response to Reviewers

2. In your Methods section, please provide additional information about the participant recruitment method and the demographic details of your participants. Please ensure you have provided sufficient details to replicate the analyses such as:

a) the recruitment date range (month and year),

b) a description of any inclusion/exclusion criteria that were applied to participant recruitment,

c) a statement as to whether your sample can be considered representative of a larger population, and

d) a description of how participants were recruited.

Authors response

Recruitment of study participants was done at the HIV clinic, well-baby and the Paediatric outpatient clinics using the systematic sampling method.

The demographic details are provided as follows; 

Age, gender, ethnicity and socioeconomic status (details are shown in page 8 of the manuscript)

a. The recruitment date range (month and year), is already provided in the manuscript (August 2017 to December 2017).

b. 

STUDY CRITERIA

Inclusion criteria for subjects (HIV infected)

1. Children aged between 2 months to 13 years who are HIV positive following the HIV screening algorithm.

2. Children whose parents/guardians gave informed consent to participate in the study; (and assent obtained for children 7years and above).

3. Those that completed the administered questionnaire properly.

Exclusion criteria for subjects (HIV infected)

• Children whose parent(s) / guardian refused or withdrew their consent.

• Those with acute febrile illness, seizure disorders and cerebral palsy.

Inclusion criteria for HIV naive (healthy controls)

1. Children aged between 2 months to 13 years with no evidence of HIV infection on a rapid screening test.

2. Children whose parents/guardians gave informed consent to participate in the study; (and assent obtained for children 7 years and above).

3. Participants who completed the administered questionnaire properly.

Exclusion criteria for HIV naive (healthy controls)

• Children whose parent(s) / guardians refused or withdrew their consent.

• Children with chronic diseases such as confirmed sickle cell anaemia, confirmed chronic liver disease, cerebral palsy and seizure disorders.

c. Yes, the sample can be considered representative of a larger population. The study sample size was determined using appropriate methodology. 

The minimum sample size was determined using the formula for calculating sample size for the comparison of two independent proportions. (28)

n per group = 2 (Zα + Zβ) 2P (1─ P) 

(Po─ P1)2

Where 

Zα = the standard normal deviate of α at 95% confidence interval, (the probability of making a type 1 error) = 1.96

Zβ = the standard normal deviate of β (the power) is set at 90% = 1.28

 Po = the prevalence of hepatitis B co-infection among HIV-infected children

A prevalence of 10% was used based on the work by Durowaye et al (29)

P1 = the prevalence of hepatitis B in children;

A prevalence of 0.5% was used based on the work by Okonko et al (30)

P is the arithmetic average of the two proportions Po+ P1

2

d. Patients were recruited using the systematic sampling method. 

• From the records in the HIV paediatric clinic of LUTH, the average monthly attendance of children was about 100 (with the clinic running once a week); therefore, an assumption of average weekly attendance of 25 children was made. A list of all registered patients for each clinic attendance was obtained from the recording officer. The 1st participant was randomly selected from the list and subsequently, every 4th number on the list was selected. An average of 12 children whose parents gave permission and/ assents was selected every clinic day from among the attendees, and this process continued until the minimum required sample size determined by appropriate sample size and representative of the larger population was achieved. For every 4th name that declined the 5th name was chosen.

• Recruitment of study participants was done at the paediatric HIV clinic of LUTH over five months (August 2017 to December 2017). The researcher on each clinic day addressed all the parents/ guardian and the patients present where details of the study were communicated to them. Consent and assent forms were given to parents or guardians and to patient aged 7 to 13 years who satisfied the inclusion criteria. Consenting parents or guardian and children 7-13 years were subsequently recruited into the study.

• The same method was used to recruit healthy controls from the well-baby clinic and paediatric outpatient unit. These children were age and sex-matched with the subjects.

3.Genital circumcision is defined as the removal of the foreskin from the penis in this manuscript.

4. Please provide the name and catalogue number of the anti-HCV Version 4.0 Immunoassay.

Response: Version 4.0 Enzyme Immunoassay; DIA.PRO Diagnostic Bioprobes Srl, Italy.) HCV Ab (DIA.PRO), Lot C11T12/5.

Comments to the Author

5.Review Comments to the Author

Reviewer #1: 1. When describing the prevalence of HIV/HBV and HIV/HBV respectively, especially in the Introduction when describing the published literature, please be clearer and describe the range prevalence for each of the viral co-infections. In the current form, it is rather unclear what the authors are referring to in their ranges of prevalence.

Response: The prevalence of HIV/HBV and HIV/HCV in the introduction has been addressed as seen in pages 4 and 5.

2. In the discussion, more can and should be said about whether there are variations in the rate of Hepatitis B vaccination between the different regions in Nigeria in which the prevalence of HIV/HBV has been studied, to account for the differences reported.

Response: There are variations in the rate of Hepatitis B vaccination between the different regions in Nigeria which has been addressed in the discussion as seen in pages 19 and 20.

3. The discussion is rather confusingly structured. Would recommend that the authors reframe it speaking of each of the viral hepatitides in sequence, without the current structure which shifts between HBV and HCV, for ease of understanding.

Response: Each viral hepatitis has been discussed in sequence in the discussion as seen in pages 18 to 22. 

4. A thorough proofreading and rewriting were needed to make it easier to read and understand would also help in the overall quality of the manuscript.

Response: It has been addressed accordingly.

---

## [Decision Letter · Decision Letter 1]

25 Nov 2020

Prevalence of and risk factors for hepatitis B and C viral co-infections in HIV infected children in Lagos, Nigeria.

PONE-D-20-23898R1

Dear Dr. Lawal,

We’re pleased to inform you that your manuscript has been judged scientifically suitable for publication and will be formally accepted for publication once it meets all outstanding technical requirements.

Kind regards,

Isabelle Chemin, PhD

Academic Editor

PLOS ONE

Additional Editor Comments (optional):

Reviewers' comments:

Reviewer's Responses to Questions

**Comments to the Author**

1. If the authors have adequately addressed your comments raised in a previous round of review and you feel that this manuscript is now acceptable for publication, you may indicate that here to bypass the “Comments to the Author” section, enter your conflict of interest statement in the “Confidential to Editor” section, and submit your "Accept" recommendation.

Reviewer #1: All comments have been addressed

2. Is the manuscript technically sound, and do the data support the conclusions?

Reviewer #1: Yes

3. Has the statistical analysis been performed appropriately and rigorously? 

Reviewer #1: Yes

4. Have the authors made all data underlying the findings in their manuscript fully available?

Reviewer #1: Yes

5. Is the manuscript presented in an intelligible fashion and written in standard English?

Reviewer #1: Yes

6. Review Comments to the Author

Reviewer #1: (No Response)

7. PLOS authors have the option to publish the peer review history of their article (what does this mean?). If published, this will include your full peer review and any attached files.

Reviewer #1: **Yes: **Wong Chen Seong

---

## [Editor Report · Acceptance letter]

1 Dec 2020

PONE-D-20-23898R1 

Prevalence of and risk factors for hepatitis B and C viral co-infections in HIV infected children in Lagos, Nigeria. 

Dear Dr. Lawal:

I'm pleased to inform you that your manuscript has been deemed suitable for publication in PLOS ONE. Congratulations! Your manuscript is now with our production department. 

Kind regards, 

on behalf of

Mrs Isabelle Chemin 

Academic Editor

PLOS ONE